# Synthesis, Characterization, and Antibacterial Assessment (Synergism) of Silver Nanoparticles Prepared with Stem Bark Extract of *Sterculia diversifolia*

Fazle Rabbi [1], Imad Ahmad [2], Amna Nisar [3], Abdur Rauf [4,*], Abdulrahman Alshammari [5], Metab Alharbi [5] and Hafiz Ansar Rasul Suleria [6]

1 Department of Pharmacy, Abasyn University Peshawar, Peshawar 25000, Khyber Pakhtunkhwa, Pakistan
2 Department of Pharmacy, Abdul Wali Khan University Mardan, Mardan 23200, Khyber Pakhtunkhwa, Pakistan
3 Department of Pharmacy, University of Peshawar, Peshawar 25120, Khyber Pakhtunkhwa, Pakistan
4 Department of Chemistry, University of Swabi, Anbar 23561, Khyber Pakhtunkhwa, Pakistan
5 Department of Pharmacology and Toxicology, College of Pharmacy, King Saud University, P.O. Box 2455, Riyadh 11451, Saudi Arabia
6 School of Agriculture and Food, Faculty of Veterinary and Agricultural Sciences, The University of Melbourne, Parkville, VIC 3010, Australia
* Correspondence: abdurrauf@uoswabi.edu.pk

**Abstract:** Microbial infections present a challenging arena to the modern world. Traditional antibiotics are now familiar to microbes. To counter this microbial familiarity, a novel approach is a nanoparticle-based drug delivery system that exhibits promising results and overcomes these problems. This study was conducted to explore the efficacy of silver nanoparticles (AgNPs) by utilizing stem bark extract of *Sterculia diversifolia* followed by physicochemical characterization including ultraviolet-visible spectrophotometry (UV-Vis), X-ray diffractometer (XRD), scanning electron microscopy (SEM), transmission electron microscopy (TEM), and Fourier-transform infrared spectroscopy. The UV-Vis characteristic spectral peak was recorded at 430 nm. XRD confirmed the crystalline structure of AgNPs, while FTIR confirmed phytochemicals in their capping, stabilization, and synthesis of AgNPs. SEM devised the particle size range of 100 nm at 30,000× magnification. TEM showed nanoparticles morphology, which is spherical in nature, while obtained nanoparticles were 100 nm in size. The antibacterial activity of synthesized NPs showed significant action against *S. aureus* and *P. aeruginosa*. Similarly, crude extract and n-hexane fraction showed maximum zone of inhibition. Promising results suggest that stem bark extract AgNPs of *Sterculia diversifolia* can be studied further for microbial mechanisms as well as formulation-based studies.

**Keywords:** green synthesis; silver nanoparticles; crude extract; antibacterial; treatment





## 1. Introduction

Green synthesis is a more reliable and economical way to synthesize metal nanoparticles. It is becoming increasingly important due to its simplicity, cost-effectiveness, stability, less time consumption, non-toxic by-products, and eco-friendliness [1]. Because of issues related to large amounts of energy consumed, the release of toxic and harmful chemicals, and the use of complex equipment and synthesis conditions, physical and chemical synthesis methods are gradually being replaced by green synthesis methods [2]. Green materials (e.g., reducing agents) contain proteins and polyphenols that can reduce metal ions ($Ag^+$) to a lower valence state, resulting in higher quality than chemically synthesized metal nanoparticles [3,4]. Polyphenols and proteins could be used as reductants to react with $Ag^+$ ions and as scaffolds to direct silver nanoparticle formation in the solution [5]. A literature survey revealed that metal nanoparticles are the most widely studied type of nanoparticles

because they are easy to synthesize. Predominantly, these NPs have a wide range of applications in medical sciences, i.e., antibacterial, targeted drug delivery, and many others. Gold, platinum, palladium, and silver are some of the commonly studied metal-based nanoparticles [6]. Silver nanoparticles (AgNPs) are one of the interesting metal-based NPs with numerous applications in health and medicine. Silver possesses strong antibacterial properties and is toxic to cells. Silver damages the bacterial cell wall, inhibits bacterial growth, and disrupts cellular metabolism as a consequence of the interaction between Ag ions with DNA and proteins inside bacterial cells [7]. This Ag ion interaction inhibits protein synthesis, reduces membrane permeability, and ultimately results in death. AgNPs, as compared to silver, are chemically more reactive. Therefore, Ag nanoparticles manifest potent antibacterial capabilities [8]. Several methods are available for the synthesis of AgNPs; an easier and more economical approach is chemical reduction [9]. Silver salt and a reducing agent (sodium borohydride or sodium citrate) are required in this method. However, other chemicals used in the process may adsorb reducing agents and organic solvents on the surface of the material, which is detrimental. Therefore, an environment-friendly method is desirable [10].

The use of medicinal plants is not restricted to the developing world. Commercially prepared herbal products are in use by developed nations as well. Worldwide, phytomedicines received a boost when the World Health Organization encouraged the use of traditional plant-based medicines to achieve better health care [11]. The practice of using plant-derived medicines has become extremely popular in the USA and Europe, with the phytomedicine industry in the USA earning $1.5 billion annually and the European market nearly three times as much [12]. The therapeutic use of about 1500 plants is widespread in European territory, including Albania, Croatia, Bulgaria, France, Germany, Hungary, Spain, Poland, and the UK. Medicinal plants are extensively utilized in Maltese island in everyday life as part of folk medicinal remedies [13]. Plant-based remedies are extensively used in the folk medicine systems of Brazil, Africa, and Asia [14].

The *Sterculiaceae* family was comprised of 60 genera and 1500 species belonging to the tropical as well as sub-tropical region [15]. This family is a source of bioactive constituents of various chemical classes, e.g., polyphenols, flavonoids, glycosides, alkaloids, steroids, terpenoids, triterpenes, saponins, sterols, apigenins, tannins, essential oils, proteins, carbohydrates, and proteins [16,17]. Medicinally, *Sterculia diversifolia* bears antibacterial, anti-glycation, anthelmintic, cytotoxic, immunomodulatory, insecticidal, larvicidal, leishmanicidal, and antioxidant activity [18–21]. Various secondary metabolites have been reported from *Sterculia diversifolia,* e.g., Gossypetin, taxifolin, methyl 4-hydroxycinnamate, and β-sitosterol-D-glucoside [18]. As mentioned above, *Sterculia diversifolia* is reported for antibacterial activity, so *Sterculia diversifolia-based* silver nanoparticles may also possess significant antibacterial potential. The literature review showed no previous reports on the investigation of the antibacterial activity of the crude extract silver nanoparticles to date. The current study will investigate the synthesis of silver nanoparticles and the antibacterial potential of crude extract AgNPs of *Sterculia diversifolia* to evaluate the plant for its folk use.

## 2. Materials and Methods

### 2.1. Materials

Blood agar, Eosin-Methylene Blue, MacConkey Agar, Mannitol Salt Agar, and Nutrient agar (Sigma Aldrich, St. Louis, MO, USA). *n*-hexane and methanol were obtained from Musaji Adam and Sons, Pakistan. Tetracycline, levofloxacin, ciprofloxacin, imipenem, amikacin, and gentamicin (Dr. Reddy's, API Suppliers, Srikakulam, India), grinder (Moulinex, Paris, France) and weighing balance (Shimadzu Corporation, Kyoto, Japan), vortex mixture (Static mixer corporation, Chicago, IL, USA), rotary evaporator (Hahnshin Scientific Co., Bucheon, South Korea).

### 2.2. Plant Collection and Taxonomic Identification

The plant was collected from the botanical garden of Pakistan Forest Institute, Peshawar, Pakistan. A specimen was deposited at the botany department herbarium, University of Peshawar, with a reference number Bot.20098, PUP. The freshly collected plant material was treated with water and dried beneath shade at room temperature for 21 days, which resulted in dried plant material. The dehydrated plant material was pulverized into powder form by means of a mechanical grinder for further processing [19,20].

### 2.3. Preparation of Extract

After grinding to powder, the plant material was macerated for two weeks with solvent (hydro methanolic: 90%), followed by filtration. The crude extract was concentrated using a rotary evaporator at 40 °C [22]. The crude extract obtained was 950 g. The crude extract was further fractionated (i.e., n-hexane, aqueous).

### 2.4. Samples collection

The urine samples were collected from UTI-positive patients from Hayatabad Medical Complex Peshawar and shifted to the Microbiology Research Laboratory of Abasyn University Peshawar. Patients were instructed regarding sample collection. The mid-stream urine was collected in the morning time using a sterilized plastic container. Samples were collected from those patients having signs and symptoms of a UTI and not on antibiotic therapy.

### 2.5. Isolation and Identification

The microorganisms were isolated from the clinical samples through pure culture techniques, including selective, differential, and complex media, i.e., Mac Conkey agar, Nutrient Agar (N.A), Mannitol Salt Agar (MSA), blood agar, and Eosin Methylene Blue (EMB). The samples were shifted to the Microbiology Laboratory of Abasyn University Peshawar in a sterile condition for preservation and further processing, such as culturing, identification, and characterization of organisms. The urine specimen was inoculated with the help of a swab in the four quadrant method on MacConkey agar plates, nutrient agar plates, blood agar, and mannitol salt agar. The streaked plates were then incubated at 37 °C for 24 h. The bacterial growth was later Gram stained. Bacterial isolates were characterized according to standard microbiological procedures. Isolates were identified through colony morphology and biochemical tests, i.e., catalase test, citrate utilization test, coagulase test, indole test, oxidase test, triple sugar iron agar, and urease test.

### 2.6. Gram Staining

Gram staining and cellular morphology of isolated bacteria were performed using a compound microscope. Smear was prepared on a clean slide by taking 24 h fresh cultures using inoculating needle, followed by drying, and then heat fixed. The smear was successively swamped with crystal violet dye, washed under tap water, application of Gram iodine for one minute, and again washed under tap water. Ethyl alcohol was used as decolorizing agent, followed by tap water washing. Then, the smear was counter-stained with safranin for one minute and washed under tap water. The prepared slides were observed under a compound microscope at 40× and 100× using immersion oil.

### 2.7. Antibacterial Activity of Crude Extracts against S. aureus and P. aeruginosa

The agar well diffusion method was used as an antibacterial assessment of *Sterculia diversifolia* extracts against MDR isolates (*Pseudomonas aeruginosa* and *Staphylococcus aureus)*. The bacterial lawn was prepared using a glass spreader on Mueller Hinton agar media plates. After the preparation of the bacterial lawn, 5 wells were bored, having 6 mm diameter, using a sterilized cork borer. Antibiotic discs were used as a positive control. Among five wells, two were used for the negative control, i.e., distilled water and DMSO, while the remaining wells were used for aqueous, n-hexane, and methanol extracts of

*Sterculia diversifolia*. A lid was used to cover the plates while edges were adhered with parafilm and then incubated at 37 °C for 24 h. A zone of inhibition was employed to assess the efficacy of *Sterculia diversifolia* extracts against isolated test organisms [23].

### 2.8. Synthesis of Silver Nanoparticles

A total of 20 mL of filtered *Sterculia diversifolia* stem bark Aqueous extract and 20 mL of 1 mM $AgNO_3$ solution were mixed in a flask to form a complete solution (pH 8.5–9.5). The mixture was centrifuged at 3000 rpm for 15 min. The precipitate was washed with distilled water to remove impurities and then kept in a dark shade for drying at 25 °C to avoid photoactivation of reduced silver ions. The dried precipitate obtained was then ground into a powder form until used [24].

### 2.9. Characterization of AgNPs

Characterization of AgNPs was carried out with UV-Visible spectroscopy (UV 1902 PC, UV-Vis Spectrophotometer, Shimadzu, Columbia, USA), SEM (JSM5910, JEOL, Tokyo, Japan), XRD (JDX 3532, Tokyo, Japan), and FTIR (IRAffinity$^{-1}$S, Shimadzu, Manchester, United Kingdom). UV-Vis spectroscopy absorption peaks in visible and ultraviolet regions of the spectrum were used to determine the characteristic $\lambda_{max}$. The determination of the concentration of an analyte in a solution follows the Beer-Lambert law, which could be carried out by specific wavelength, calculating the absorbance. In SEM analysis, a thin film of the sample was prepared by using a carbon-coated copper grid. Extra water was absorbed by blotting paper, and the film was allowed to dry under a mercury lamp for 5 min over the SEM grid. TEM was operated with an acceleration voltage of 100 kV, and the films on the TEM grids were allowed to stand for 3 min. The extra solution was removed, and the grid was allowed to dry before measurement. FTIR spectrum was used to recognize different functional groups, while X-ray diffraction determined the crystalline or amorphous nature of AgNPs.

### 2.10. Preparation of AgNP-Coated Antibiotic Discs

A stock solution of AgNPs (20 µg/µL) was prepared by dissolving 20 mg of AgNPs residue per mL of distilled water. A total of 5 µL (100 µg AgNPs) from the stock solution was poured on antibiotic discs within Petri plates and then allowed to dry at 50 °C for 20 min. The method of AgNPs coating was repeated for each type of antibiotic disc [25].

### 2.11. Disc Diffusion Assay

Both AgNPs coated and uncoated antibiotics were assessed against MDR bacterial isolates using a standard Kirby Bauer disc diffusion assay [26]. The bacterial lawn was prepared on Muller Hinton agar plates, followed by the application of AgNPs coated and uncoated antibiotic discs and then incubation at 37 °C for 24 h. After incubation, the potency of AgNPs coated and uncoated antibiotics were determined from the zone of inhibition.

### 2.12. Ethical Statement

Ethical approval of the study was obtained from the ethical committee of HMC, Peshawar, Khyber Pakhtunkhwa, Pakistan, under reference No. 47/EC-21/HMC dated 16 September 2021.

### 2.13. Statistical Analysis

GraphPad Prism V.6.0 was used for statistical analysis. Mean and S.E.M. was implemented, and statistically, $p < 0.05$ values were taken into consideration as significant. The analysis of variance was used to determine statistical significance, followed by a control group with an experimental group.

## 3. Results

### 3.1. Synthesis of Silver Nanoparticles

The initial color of the *Sterculia diversifolia* aqueous stem bark extract was changed from light green to yellowish-brown on mixing with 2 mM $AgNO_3$ solution. However, no further change in color was noted after 24 h of incubation. This indicates a high concentration of flavonoids, phenols, and other phytochemicals may be involved in the bio-reduction of Ag to AgNPs.

### 3.2. Characterization of Nanoparticles

#### 3.2.1. UV-Vis Spectroscopic Confirmation

Upon UV-Visible spectroscopy, AgNPs characteristic peak was obtained at 430 nm, whereas silver nitrate and aqueous fraction did not show any characteristic peaks, as shown in Figure 1.

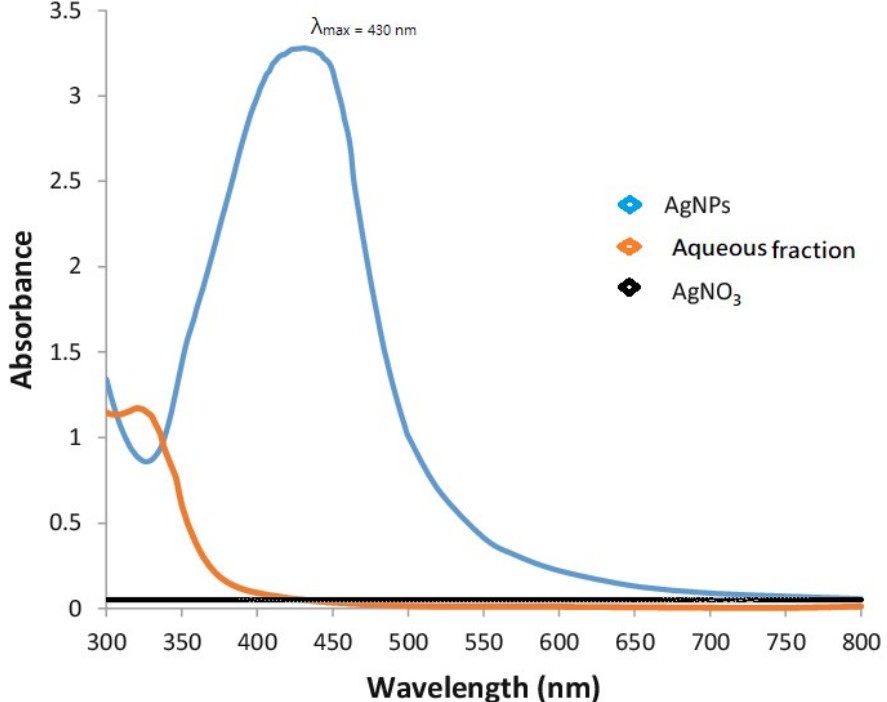

**Figure 1.** UV-visible spectra of synthesized AgNPs showed a highly intensive peak of silver. A strong peak at 430 nm is evident.

#### 3.2.2. Scanning Electron Microscopy and Transmission Electron Microscopy

SEM micrograph confirmed mono-dispersed and irregular morphology of AgNPs, with 100 nm particle size at a magnification of $30,000\times$. SEM declared *Sterculia diversifolia* aqueous extract as a strong reducing agent and resulted in irregularly shaped AgNPs, as shown in Figure 2a.

TEM image of silver nanoparticles derived from *Sterculia diversifolia* aqueous extract was shown in Figure 2b. These nanoparticles' morphology was spherical in nature. The obtained nanoparticles were 100 nm in size.

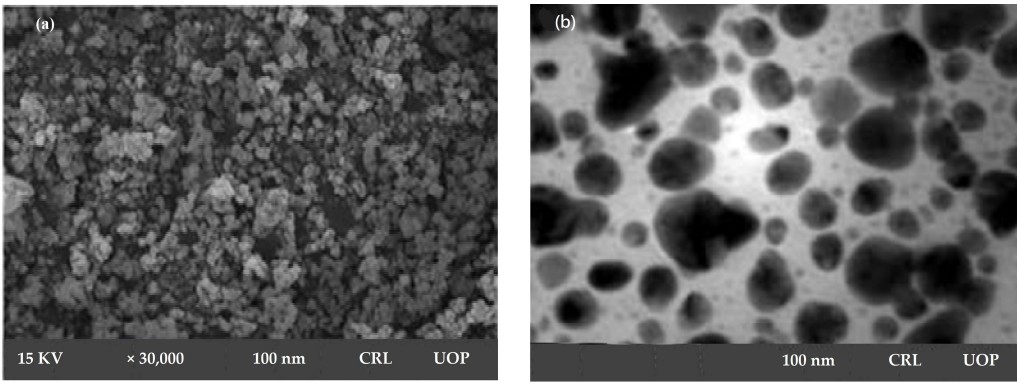

**Figure 2.** SEM micrograph (**a**) and TEM micrograph (**b**) of synthesized AgNPs.

### 3.2.3. X-ray Diffraction Method

XRD spectrum revealed that the sample was a finely grounded and homogenized substance. Similarly, XRD analysis declared the crystal-like nature of AgNPs (Figure 3). A total of 4 distinct diffraction peaks obtained at 2θ values of 38.24, 44.32, 64.48, and 77.44 could be indexed to the (823), (293), (223), and (337) reflection planes of cubic structure. Additional peaks other than Bragg peaks were observed at 23.64 and 32.32 θ, which were indicative of organic contaminants.

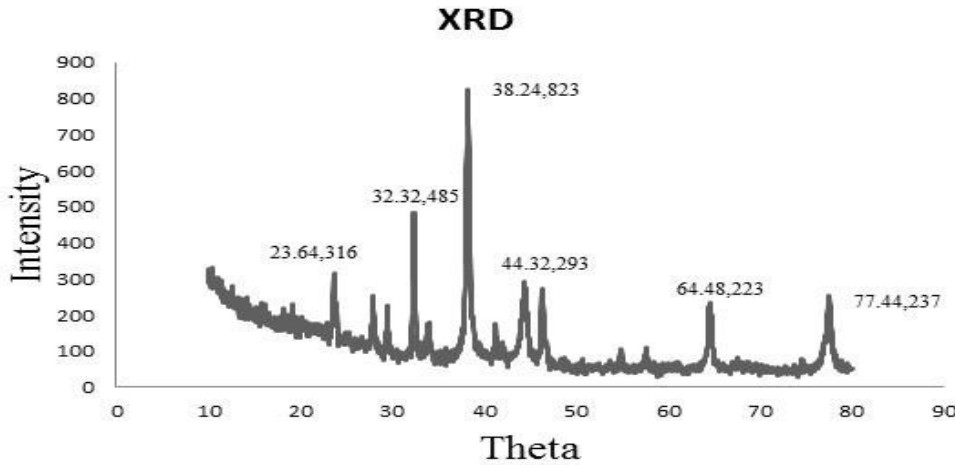

**Figure 3.** XRD spectra of synthesized AgNPs defining their crystal-like nature.

### 3.2.4. FTIR Analysis

For further characterization, FTIR analysis of an aqueous extract of *S. diversifolia* and synthesized AgNPs was carried out, and different peaks of the FTIR spectrum were compared, representing amines, carboxylic acids, and alkanes. These imperative functional groups are vital for the synthesis, capping, and stabilization of AgNPs. Peaks in the FTIR spectrum of aqueous extract of *S. diversifolia* (control) at 3411.5, 2932.6, 1749, 1637.6, 1386.5, 1146.5, 1077, 829.5, and 642.4 cm$^{-1}$ showed interrelation of different functional groups such as alcohol, alkanes, ester, amide, alkanes, carboxylic acid, aliphatic amines, or phenol and amines, respectively. After reacting with silver nitrate, the new peaks were detected at 3420.8, 2927.7, 1742.9, 1626, 1383.3, 1141.1, 1076.3, 824.5, and 651.3 cm$^{-1}$, indicating carboxylic, hydroxyl, and amide groups within the aqueous extract of *S. diversifolia*, which played a role in the synthesis of AgNPs, as shown in Figure 4.

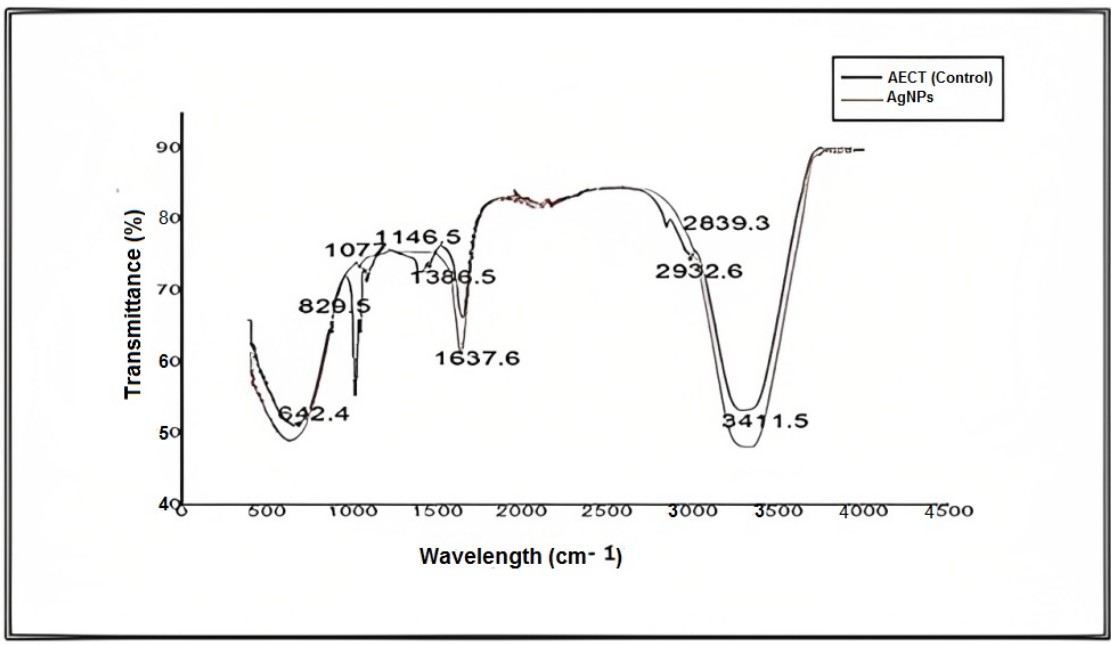

**Figure 4.** FTIR analysis of an aqueous extract of *S. diversifolia* and synthesized AgNPs.

### 3.3. Isolation and Identification of Bacterial Species

The number of isolated species of *Pseudomonas aeruginosa* and *Staphylococcus aureus* from UTI patients was ten and twenty, respectively, as shown in Table 1. Table 2 summarizes the biochemical and morphological characteristics of the bacterial species identified. *S. aureus* culture was characterized by golden yellow, circular, soft, convex, and shiny appearance. Further testing identified it as Gram-positive, catalase-positive, Lac AG, dex-negative, cit-positive, and negative, urea-positive, and TSI slant-negative. *P. aeruginosa* culture was characterized by greenish blue, low convex, and shiny rod-like appearance. Further testing identified it as Gram-negative, catalase-positive, Lac-negative, dex-negative, cit-positive, and negative, urea-positive, and TSI slant-positive.

**Table 1.** Percentage-wise distribution of bacterial species *Pseudomonas aeruginosa*, and *Staphylococcus aureus* isolated from UTI patients.

| Sr. No. | Identified Species | No. of Isolated Species (n) | Percentage |
|---------|--------------------|-----------------------------|------------|
| 1 | *P. aeruginosa* | 10 | 33.3% |
| 2 | *S. aureus* | 20 | 66.7% |

**Table 2.** Bacterial species are identified based on biochemical and morphological characteristics.

| Sr. No. | Culture Characteristics | Gram rxn | Cat | Lac | Dex | Cit | Ind | Urea | TSI Slant/Butt | Species Identified |
|---------|-------------------------|----------|-----|-----|-----|-----|-----|------|----------------|--------------------|
| 1 | Golden yellow, circular, soft, convex, shiny | + | + | AG | − | + | − | + | − | *S. aureus* |
| 2 | Greenish blue low convex shiny, rod | − | + | − | − | + | − | + | + | *P. aeruginosa* |

+ = Positive reaction; − = Negative reaction; d = Variable reaction; Cat = Catalase; lac = lactase; Dex = Dextrose; Cit = Citrate utilization; Ind = Indole production; Urea = Urease production, TSI = Triple sugar iron; A = Acidic; NA= Not applicable; A/NC = Acid/no color change.

### 3.4. Antibacterial Activity of AgNPs

The action of coated AgNPs and an uncoated antibiotic against *P. aeruginosa* and *S. aureus* test isolates are given in Table 3 and Figure 5. Against *S. aureus*, the highest potency was exhibited by tetracycline (62.5%), followed by levofloxacin (38.8%), meropenem (37.5%),

gentamicin (36.3%), ciprofloxacin (22.2%), amikacin (19%), and imipenem (12%). In the case of *P. aeruginosa*, tetracycline demonstrated 100% potency, followed by meropenem (33.3%), gentamicin (30.7%), ciprofloxacin (18.5%), imipenem (18.2%), levofloxacin (15%), and amikacin (15%).

**Table 3.** The activity of Coated and uncoated Antibiotic against test *S. aureus* and *P. aeruginosa* Isolates.

| Antibiotics Activity Measured in mm | | *S. aureus* | *P. aeruginosa* |
|---|---|---|---|
| Tetracycline | Uncoated | 8 mm | 1 mm |
| | AgNPs coated | 13 mm | 11 mm |
| | Inc. % Potency | 62.5% | 100% |
| Levofloxacin | Uncoated | 18 mm | 20 mm |
| | AgNPs coated | 25 mm | 23 mm |
| | Inc. % Potency | 38.8% | 15% |
| Meropenem | Uncoated | 8 mm | 6 mm |
| | AgNPs coated | 11 mm | 8 mm |
| | Inc. % Potency | 37.5% | 33.3% |
| Gentamicin | Uncoated | 11 mm | 13 mm |
| | AgNPs coated | 15 mm | 17 mm |
| | Inc. % Potency | 36.3% | 30.7% |
| Ciprofloxacin | Uncoated | 18 mm | 19 mm |
| | AgNPs coated | 22 mm | 22 mm |
| | Inc. % Potency | 22.2% | 15.8% |
| Amikacin | Uncoated | 21 mm | 20 mm |
| | AgNPs coated | 25 mm | 23 mm |
| | Inc. % Potency | 19.0% | 15.0% |
| Imipenem | Uncoated | 25 mm | 22 mm |
| | AgNPs coated | 38 mm | 26 mm |
| | Inc. % Potency | 12.0% | 18.2% |

AgNPs = Silver nanoparticles; mm = Millimeter.

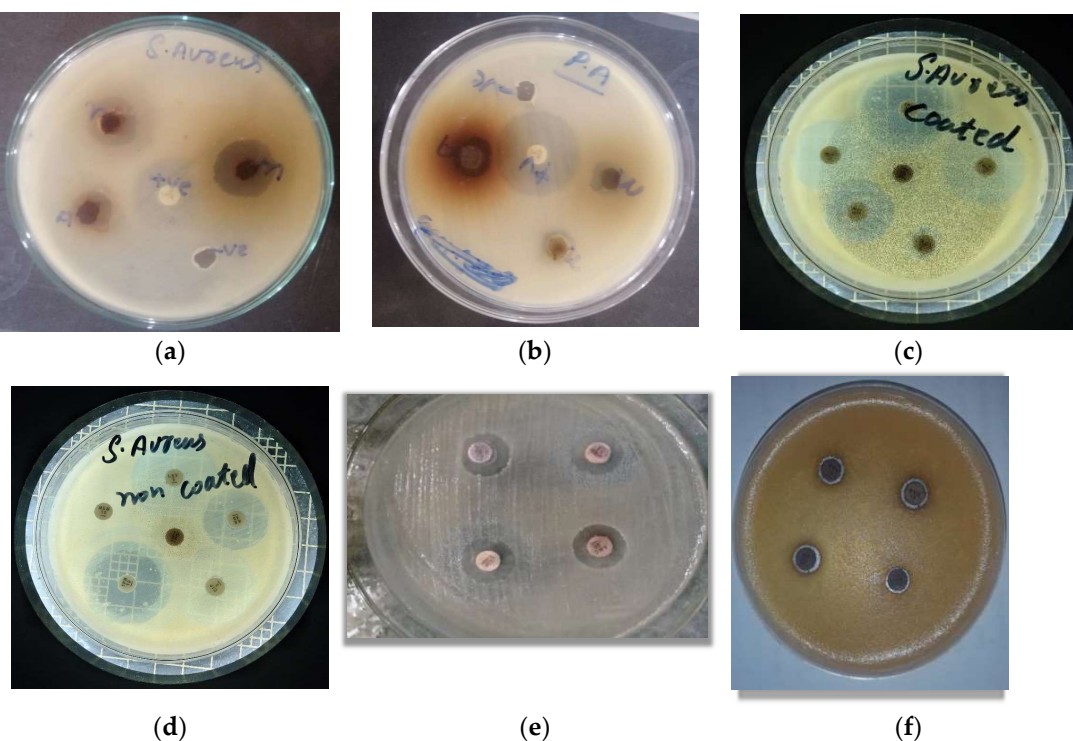

**Figure 5.** The activity of coated and uncoated antibiotic against test *S. aureus* and *P. aeruginosa* isolates; (**a**) Activity of AgNPs against *S. aureus* (**b**) Activity of AgNPs against *P. aeruginosa* (**c**) Activity of AgNPs coated Antibiotics against *S. aureus* (**d**) Activity of non-coated Antibiotics against *S. aureus* (**e**) Activity of AgNPs coated Antibiotics against *P. aeruginosa* (**f**) Activity of AgNPs non-coated Antibiotics against *P. aeruginosa*.

### 3.5. Anti-Bacterial Activity of S. diversifolia Stem Bark Crude Extract against Staphylococcus aureus

Methanol crude extract showed 22 mm, 20 mm, and 23 mm inhibition zone against *S. aureus* isolate 1, 2, and 3, while n-hexane crude extract showed 17 mm, 16 mm, and 19 mm inhibition zone against *S. aureus* isolates 1 to isolate 3, respectively. The aqueous crude extract of *S. diversifolia* revealed 10 mm, 11 mm, and 13 mm inhibition zones against *S. aureus* isolate 1 to isolate 3, respectively, while ciprofloxacin was used as a positive control, which showed 22 mm, 24 mm, and 25 mm inhibition zone against isolate 1 to 3, respectively. The negative control (DMSO) showed no antibacterial activity against any isolate of *S. aureus,* as shown in Figure 6.

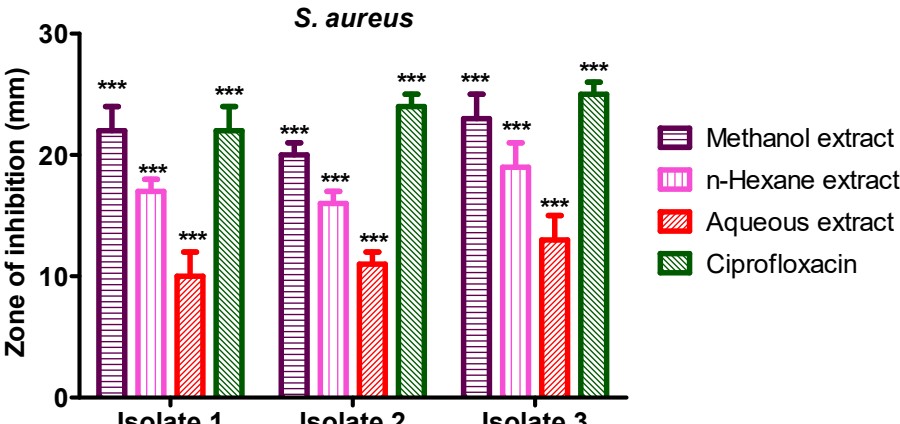

**Figure 6.** Antibacterial activity of *S. diversifolia* extracts against *S. aureus* at 100 μL concentration. ANOVA was applied before Dennett's post hoc analysis. *** $p < 0.001$ compared to negative control group.

*3.6. Anti-Bacterial Activity of S. diversifolia Stem Crude Extract against P. aeruginosa*

Methanol crude extract showed 22 mm, 23 mm, and 21 mm inhibition zones against *P. aeruginosa* isolate 1, 2, and 3, while *n*-hexane crude extract showed 20 mm, 21 mm, and 19 mm inhibition zone against *P. aeruginosa* isolates 1 to 3, respectively. The aqueous crude extract of *S. diversifolia* revealed 12 mm, 14 mm, and 15 mm inhibition zones against *P. aeruginosa* isolate 1 to isolate 3, respectively. Ciprofloxacin was used as a positive control, which showed 23 mm, 24 mm, and 22 mm inhibition zones against isolate 1 to 3, respectively. The negative control (DMSO) showed no antibacterial activity against any isolate of *P. aeruginosa,* as shown in Figure 7.

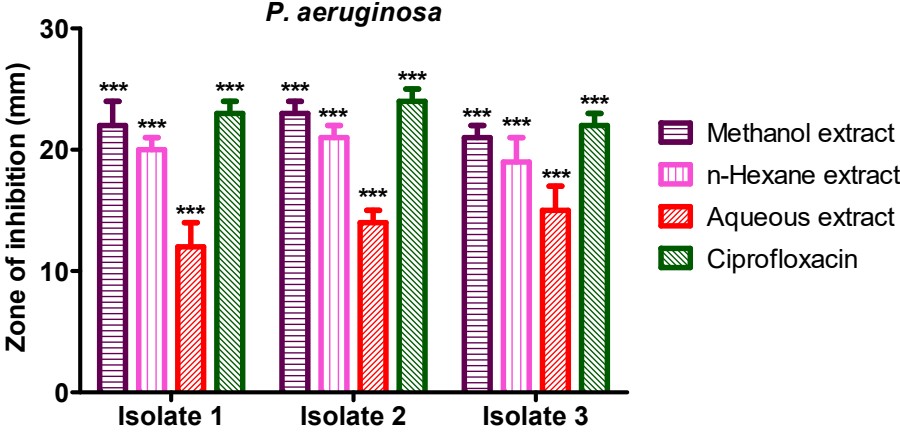

**Figure 7.** Antibacterial activities of *S. diversifolia* extract against *P. aeruginosa* (100 μL). ANOVA was applied before Dennett's post hoc analysis. *** $p < 0.001$ compared to negative control group.

## 4. Discussion

MDR bacterial infections are very common all over the world, threatening public health and treatment strategies [27]. The development of new antibacterial agents, along with the safe use of antibiotics and other antimicrobial agents, played an important role in controlling antibiotic resistance. Discouraging empirical antibiotic therapy and optimized treatment duration could help reduce antibiotic resistance phenomena [28]. Methicillin-Resistant *Staphylococcus aureus* (MRSA) is one such strain of *S. aureus* that has developed resistance to β-lactam antibiotics, including methicillin and oxacillin. The market available vancomycin is still a magic bullet against MRSA [29]. However, resistance is more prevalent in different localities against this magic bullet [30,31].

In the same manner, *Pseudomonas aeruginosa* is a Gram-negative bacterium infecting humans, especially immune-compromised patients. *P. aeruginosa* has demonstrated resistance against different antibiotics belonging to the β-lactam group, even aminoglycosides and quinolones [32]. The emergence of antibiotic-resistant strains, and more fatal, is extensively drug resistance (XDR) [33,34]. Apart from conventional antibiotics, another way to counter this global resistance is the use of an advanced drug delivery system. One of which is a nanotechnology-based approach that relies on small, sophisticated particles of <100 nm size to achieve a better drug response [35]. Amongst various modalities of nanoparticles targeted towards microbes, silver-containing formulations are common. However, the nano approach toward safe, efficacious, potent, and targeted formulation is still in the pooling stage. From this reservoir, one of the hit formulations will be picked up [36].

Amongst numerous approaches for nanoparticle synthesis, a nature-based approach known as the biological method is considered to be safe and efficacious as compared to chemical and physical techniques. Biological synthesis incorporates naturally occurring reducing agents, i.e., bacteria, fungi, plant extracts, yeast, enzymes, proteins, peptides, polysaccharides, etc. Bio-inspired green synthesis of silver nanoparticles using plant extract is considered a reliable method [36–40].

During this study, the antibacterial activity of synthesized NPs showed significant action against *S. aureus* and *P. aeruginosa,* which is almost comparable to the literature. AgNPs possess outstanding bactericidal properties against a wide variety of pathogenic bacterial species [41]. The application of AgNPs in catalysis, drug administration, antimicrobial activities, electronics, and different biological systems make them eco-friendly [42]. The biosynthesis of AgNPs utilizes actinomycetes, bacteria, fungi, and different parts of medicinal plants, such as flowers, leaves, and fruits [43]. The size, shape, and stability of AgNPs largely depend on the method of synthesis and the temperature of the manufacturing process [44].

Our synthesized AgNPs were monodispersed with irregular 100 nm particle size morphology. This is an ideal size for nanoparticles used for human/medical use, as reported in the literature [45,46]. Plant extract-based NPs are free from pollutants, with a defined size and morphology as compared to chemically synthesized nanoparticles [46,47]. That is the reason we selected the green synthetic approach, and it was fruitful, evident from our results. Characterization via FTIR analysis identified newly formed functional groups of carboxylic acid, hydroxy, amide, etc. These functional groups were considered important for our AgNPs and congruent with the available literature [48].

The cubic crystallographic nature of our AgNPs is another ideologic feature [49]. Cubic nanoparticles offer a large contact area between the nanoparticle and the reference surface [50]. This may increase the pharmacological action of these nanoparticles, producing more potent action. As in our experiments, the antibacterial action of synthesized AgNPs has been increased. The surface-volume ratio, as well as the curvature dimension of nanoparticles, will determine their cellular behavior [51]. The formation of cubic nanoparticles enables them for better biodistribution and cellular uptake or internalization. Such features make these types of nanoparticles achieve unmet clinical needs. Due to this high aspect ratio, cubic nanoparticles achieve a larger extent and faster rate of absorption than other lower aspect ratio-carrying nanoparticles (spherical) [52].

Against *S. aureus*, the potency of tetracycline was increased by 62.5%. It is worth to be noticed that the potency of protein synthesis inhibitors is increased by our AgNPs. Therefore, the possible mechanism looks to be synergistic action but is subject to confirmation in the lab. In the case of *P. aeruginosa*, the same synergistic effect by tetracycline demonstrated 100% potency amongst all tested antibiotics. Therefore, it seems that our hypothesis is probably strong. It is also notable that the potency of all antibiotics was potentiated by our synthesized AgNPs.

## 5. Conclusions

In the current study, green synthesis (eco-friendly and non-toxic) of AgNPs was achieved using plant extract of *Sterculia diversifolia* that act as a reducing agent. The synthesis of AgNPs was characterized by different techniques, such as UV-Visible, FT-IR, SEM, TEM, and XRD analysis. The green synthesized AgNPs were used to evaluate the antibacterial activity against *S. aureus* and *P. aeruginosa,* showing promising activity in comparison to positive control. Our study provides preliminary data that AgNPs can inhibit bacterial growth. However, further studies are required to show the efficacy of AgNPs.

**Author Contributions:** Conceptualization, F.R.; methodology, I.A.; investigation, A.N.; writing-original draft preparation, A.R.; writing, review, and editing, A.A., M.A., H.A.R.S. All authors have read and agreed to the published version of the manuscript.

**Funding:** All the authors are thankful to the Researchers Supporting Project number (RSP2023R491), King Saud University, Riyadh, Saudi Arabia.

**Data Availability Statement:** All authors have confirmed that no new data were created.

**Acknowledgments:** All the authors are thankful to the Researchers Supporting Project number (RSP2023R491), King Saud University, Riyadh, Saudi Arabia.

**Conflicts of Interest:** The authors declare no conflict of interest.

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
