# Peer review of "Synthesis, Characterization, and Antibacterial Assessment (Synergism) of Silver Nanoparticles Prepared with Stem Bark Extract of Sterculia diversifolia"

_crystals, doi:10.3390/cryst13030480_

Round 1
Reviewer 1 Report
The manuscript illustrates plant-mediated AgNPs antibacterial activity. The authors made an effort to isolate bacteria from UTI patients. The ethical approval for taking clinical urine samples from patients is not mentioned.
The most common cause of UTI is E. coli, and that data was missing.
The isolated microorganism according to the authors is MDRS. But it is made clear through experiments. The antimicrobial activity of AgNPs is widely established.
Therefore using the antibiotic disc in combination with plant-mediated AgNPs, makes little sense If not, the authors must demonstrate that Plant mediated AgNPs alone exhibited high antimicrobial efficacy compared to antibiotic.
Overall the article needs additional data before publication.
Author Response
Reviewer 1
- The manuscript illustrates plant mediated AgNPs antibacterial activity. The authors made an effort to isolate bacteria from UTI patients. The ethical approval for taking clinical urine samples from patients is not mentioned.
Response: Ethical approval of the study was obtained from the ethical committee of HMC, Peshawar, Khyber Pakhtunkhwa, Pakistan under the reference No. 47/EC-21/HMC dated 16 September 2021.
- The most common cause of UTI is E. coli, and that data was missing.
Response: We have selected only species of pseudomonas aeruginosa and Staphylococcus aureus, because these resistant microbes are prevalent in our region. This is supported from literature.
- The isolated microorganism according to the authors is MDRS. But it is made clear through experiments. The antimicrobial activity of AgNPs is widely established. Therefore, using the antibiotic disc in combination with plant mediated AgNPs, makes little sense If not, the authors must demonstrate that Plant mediated AgNPs alone exhibited high antimicrobial efficacy compared to antibiotic.
Response: Using an antibiotic disc in combination with plant mediated AgNPs is justified by our research title. It is also noteworthy that antibiotics and AgNPs were tested individually at first, and that the antibacterial activity of Plant mediated AgNPs is evident upon comparison of antibiotic alone and in combination with AgNPs.
- Overall, the article needs additional data before publication.
Response: Respected reviewer, we have incorporated relevant additions (green synthesis, reduction mechanism, hypothesis, TEM, discussion), revisions, and references as per reviewer’s comments.
Reviewer 2 Report
The manuscript contains valuable information. However, it needs some modifications as follows:
1. The data presented in Figure 6 should be analyzed statistically at the confidence interval of 95%.
2. The antimicrobial activity of phytosynthesized silver nanoparticles in this study should be compared with similar studies such as the following studies: Bioengineering of green-synthesized silver nanoparticles: In vitro physicochemical, antibacterial, biofilm inhibitory, anticoagulant, and antioxidant performance; Penicillium chrysogenum-derived silver nanoparticles: exploration of their antibacterial and biofilm inhibitory activity against the standard and pathogenic Acinetobacter baumannii compared to tetracycline.
3. In the introduction describe the "Green synthesis" and compare it with other methods of NPs synthesis.
4. The hypothesis for the synthesis is lacking in the introduction.
5. The author should provide the reduction mechanism of nanoparticle synthesis in the presence of plant extract and specify at least some compounds which are the reason for the reduction.
6. Please provide the state-of-art. On the other hand, why such nanoparticles are useful and selected? What is the knowledge gap and novelty of the work?
7. In the section of Discussion, the authors should also mention the factors that may have an influence on the biological activity of inorganic nanoparticles. These factors include size distribution, morphology, surface charge, surface chemistry, capping agents, etc.
8. The authors are suggested to mention the pharmaceutical and biological potential of green synthesized silver nanoparticles such as antiviral, antibacterial, antifungal, antiparasitic (Reference), antioxidant, anticoagulant, and biofilm inhibitory activities of biosynthesized silver nanoparticles and also the anticancer activity of biosynthesized silver nanoparticles against different cancers such as lung, colorectal, leukemia, hepatic, etc.
Author Response
Reviewer 2
- The data presented in Figure 6 should be analyzed statistically at the confidence interval of 95%.
Response: Actually ANOVA was applied for statistical analysis, before Dennett's post hoc analysis. ***p < 0.001, **p < 0.01, *p < 0.05 (confidence interval 95%) compared to negative control group.
- The antimicrobial activity of phytosynthesized silver nanoparticles in this study should be compared with similar studies such as the following studies: Bioengineering of green-synthesized silver nanoparticles: In vitro physicochemical, antibacterial, biofilm inhibitory, anticoagulant, and antioxidant performance; Penicillium chrysogenum-derived silver nanoparticles: exploration of their antibacterial and biofilm inhibitory activity against the standard and pathogenic Acinetobacter baumannii compared to tetracycline.
Response: The antibacterial activity of phytosynthesized silver nanoparticles in this study has been compared in the discussion portion with similar studies as mentioned in the literature.
(Talank, N.; Morad, H.; Barabadi, H.; Mojab, F.; Amidi, S.; Kobarfard, F.; Mahjoub, M. A.; Jounaki, K.; Mohammadi, N.; Salehi, G. Bioengineering of green-synthesized silver nanoparticles: In vitro physicochemical, antibacterial, biofilm inhibitory, anticoagulant, and antioxidant performance. Talanta 2022, 243, 123374.)
- In the introduction describe the "Green synthesis" and compare it with other methods of NPs synthesis.
Response: Synthesis of nanoparticles is discussed in discussion. However Green synthesis has been described in introduction and also compared with chemical and physical synthesis methods of nanoparticles.
- The hypothesis for the synthesis is lacking in the introduction.
Response: Sterculia diversifolia is reported for antibacterial activity, but Sterculia diversifolia based silver nanoparticles with antibacterial properties could be the hypothesis and added to the introduction portion as well.
- The author should provide the reduction mechanism of nanoparticle synthesis in the presence of plant extract and specify at least some compounds which are the reason for the reduction.
Response: The reduction mechanism of nanoparticle synthesis in the presence of plant extract has been mentioned in introduction section. As we know green materials (e.g. reducing agents) contain proteins and polyphenols that can reduce metal ions (Ag+) to a lower valence state. Polyphenols and proteins could be used as reductants to react with Ag+ ions and as scaffolds to direct formation of silver nanoparticles in solution.
- Please provide the state-of-art. On the other hand, why such nanoparticles are useful and selected? What is the knowledge gap and novelty of the work?
Response: Sterculia diversifolia has proven antibacterial activity, and there is no report on its synthesised silver nanoparticle antibacterial assay. Therefore, it is evident that our study has identified novel sterculia diversifolia based nanoparticles with a significant antibacterial action.
- In the section of Discussion, the authors should also mention the factors that may have an influence on the biological activity of inorganic nanoparticles. These factors include size distribution, morphology, surface charge, surface chemistry, capping agents, etc.
Response: Factors affecting nanoparticles are discussed in the discussion portion, keeping in view the title and theme of this article, e.g. morphology, crystallography, size distribution, surface-volume ratio and their interrelationship with absorption, biodistribution and pharmacological activity. These selective properties are discussed keeping in view the characterization techniques used in this research.
- The authors are suggested to mention the pharmaceutical and biological potential of green synthesized silver nanoparticles such as antiviral, antibacterial, antifungal, antiparasitic (Reference), antioxidant, anticoagulant, and biofilm inhibitory activities of biosynthesized silver nanoparticles and also the anticancer activity of biosynthesized silver nanoparticles against different cancers such as lung, colorectal, leukemia, hepatic, etc.
Response: As the article is focused on antibacterial activity of silver nanoparticals, therefore, other biological activities seem non-relevant.

Reviewer 3 Report
I had the pleasure to read this manuscript which presented fabrication and antibacterial performance of Ag NPs based on stem bark extract of Sterculia diversifolia. The work seems to be interesting. However, before accepting the paper, I suggest some revisions addressing the following comments.
1. In the manuscript, “antimicrobial” was suggested to be replaced with a common term “antibacterial”.
2. The material characterizations were relatively simple. The authors are suggested to add TEM/HRTEM and also XPS results of the as-synthesized Ag nanoparticles.
3. All the figures need to be revised with neat and consistent layouts. The readability of figures should be improved with the standard of academic journal. In Figure 6 and 7, the results of DMSO are suggested to be deleted or demonstrated in a more intuitive way.
4. For antibacterial evaluation, spread plate and colony counting may be more intuitive and common. The authors need to explain the reason for choosing the diverse method. Besides, some typical spread plate tests can be added for better demonstration.
5. The authors are suggested to add a section of “conclusions”. This may be not mandatory, but it is typically believe to be useful.
6. The ref 8 should be checked with format. The following references can be added in the manuscript, which include recent advances in micro/nanofabrication and biotemplated synthesis (DOI: 10.1002/sstr.202200356; 10.1021/acsami.1c16859).
Author Response
Reviewer 3
- In the manuscript, “antimicrobial” was suggested to be replaced with a common term “antibacterial”.
Response: “Antimicrobial” was replaced with “antibacterial” term as suggested
- The material characterizations were relatively simple. The authors are suggested to add TEM/HRTEM and also XPS results of the as-synthesized Ag nanoparticles.
Response: TEM results was added as per reviewer’s suggession
- All the figures need to be revised with neat and consistent layouts. The readability of figures should be improved with the standard of academic journal. In Figure 6 and 7, the results of DMSO are suggested to be deleted or demonstrated in a more intuitive way.
Response: Revised as well as deleted DMSO results as suggested
- For antibacterial evaluation, spread plate and colony counting may be more intuitive and common. The authors need to explain the reason for choosing the diverse method. Besides, some typical spread plate tests can be added for better demonstration.
Response: This protocol is commonly followed in our laboratory that is the main reason for choosing this method.
- The authors are suggested to add a section of “conclusions”. This may be not mandatory, but it is typically believe to be useful.
Response: Section of conclusions have been added as suggested by the reviewer.
- The ref 8 should be checked with format. The following references can be added in the manuscript, which include recent advances in micro/nanofabrication and biotemplated synthesis (DOI: 10.1002/sstr.202200356; 10.1021/acsami.1c16859).
Response: Reference # 8 has been corrected and also added one of the mentioned reference (DOI: 10.1002/sstr.202200356) as well.

Round 2
Reviewer 1 Report
This manuscript now highlights the effectiveness of the green synthesis of Silver nanoparticles and their antibacterial activity. After including the additional data now it seems to be good.
Reviewer 3 Report
I‘m satisfied with your revision.